Pashto script and graphics detection in camera captured Pashto document images using deep learning model

Bahadar Khan 1
Ahmad Riaz 1
http://orcid.org/0000-0003-3647-8578 Aurangzeb Khursheed 2
Muhammad Siraj 1
Ullah Khalil 3
Hussain Ibrar 1
Syed Ikram 4 ikram@gachon.ac.kr
http://orcid.org/0000-0001-8093-6690 Shahid Anwar Muhammad 4 Shahidanwar786@gachon.ac.kr
1 Department of Computer Science, Shaheed Benazir Bhutto University , Sheringal , Pakistan
2 Department of Computer Engineering, College of Computer and Information Sciences, King Saud University , Riyadh , Saudi Arabia
3 Department of Software Engineering, University of Malakand , Chakdara , Pakistan
4 Department of AI and Software, Gachon University , Seongnam-si , Republic of South Korea
Zhang Yu-Dong
Electronic publication date: 2024 Jul 26
Publication date: 2024
Volume: 10
Electronic Location ID: e2089
Received 2023 Nov 22; Accepted 2024 May 6
Copyright: © 2024 Bahadar et al.
Copyright year: 2024
Copyright holder: Bahadar et al.
License: This is an open access article distributed under the terms of the Creative Commons Attribution License, which permits unrestricted use, distribution, reproduction and adaptation in any medium and for any purpose provided that it is properly attributed. For attribution, the original author(s), title, publication source (PeerJ Computer Science) and either DOI or URL of the article must be cited.
License URL: https://creativecommons.org/licenses/by/4.0/

Keywords: Script detection, Graphic detection, Deep learning, Document images

Funding: King Saud University RSPD2024R947 This research is funded by Researchers Supporting Project Number (RSPD2024R947), King Saud University, Riyadh, Saudi Arabia. The funders had no role in study design, data collection and analysis, decision to publish, or preparation of the manuscript.

==============================
Layout analysis is the main component of a typical Document Image Analysis (DIA) system and plays an important role in pre-processing. However, regarding the Pashto language, the document images have not been explored so far. This research, for the first time, examines Pashto text along with graphics and proposes a deep learning-based classifier that can detect Pashto text and graphics per document. Another notable contribution of this research is the creation of a real dataset, which contains more than 1,000 images of the Pashto documents captured by a camera. For this dataset, we applied the convolution neural network (CNN) following a deep learning technique. Our intended method is based on the development of the advanced and classical variant of Faster R-CNN called Single-Shot Detector (SSD). The evaluation was performed by examining the 300 images from the test set. Through this way, we achieved a mean average precision (mAP) of 84.90%.

Introduction

Document images are referred to as digital images that acquired from scanner or camera. Such documents contain articles, postal addresses, forms, topographic maps, engineering drawings, license plates, billboards, subtitles in videos and photos (Kalaskar & Dhore, 2012). The main sources for the acquisition of these images are scanners and cameras. These document images are in the form of pixels data, and cannot be read or searched on computers directly (Dixit & Shirdhonkar, 2015). Moreover, when stored in computers, they occupy more space and thus present a challenge in term of space.

To convert these document images into machine-readable form, we need a sophisticated application for analyzing document images. Document Image Analysis (DIA) is an area of Artificial Intelligence (AI) that effectively deals with document images and their concerned applications. The main objectives of a DIA system are to recognize, extract, convert the text/ graphics segments of document images, and use these extracted segments further for several applications (O’Gorman & Kasturi, 1995). Figure 1 illustrates a typical DIA system. The major stages of the DIA system are ( 1) image acquisition, ( 2) pre-processing, ( 3) layout analysis and classification, ( 4) optical character recognition (OCR), and ( 5) OCR’s output. The method of obtaining an electronic image of a paper-based document is known as image acquisition. The basic two-dimensional image is either monochrome (gray-scale) or RGB (Balamurugan, Sangeetha & Senguttovelan, 2011). The general purpose of Image Acquisition is to transform the real-world data (visual image) into an array of digital data that can be later managed on a machine.

Figure 1 The generic DIA operating framework.

Through the pre-processing of a DIA system, an input image transforms into an improved image most appropriate for more analysis. The purpose of a pre-processing is a dismissal of the image data that hold unwanted distortions or to improve any image features essential for further processing. Moreover, geometric conversions of images (e.g., rotation, translation, and scaling) are also classified under pre-processing procedures (Marinai, 2008). Different pre-processing operations might includes binarization, noise removal, correction of perspective distortion, skew correction, etc.

Document layout analysis is the essential steps in getting the document image into electronic form. The document layout analysis recognizes important sections of a document like identification and segmentation of the number of text lines, words, segmentation of non-textual regions such as graphs, tables, charts, figures, etc., (Dhandra & Malemath, 2005). This research also focuses on the Pashto document images, and analyses the very basic layouts regarding the DIA system. There is very little work that has been addressing the Pashto language with respect to OCR for text-line recognition (Ahmad et al., 2017). However, regarding the different layouts, the Pashto document images have not been explored so far. Such layouts present the documents that contain the Pashto text and graphics.

OCR is a well-known product of the DIA system. It is software that recognizes characters in a document image. The OCR system reads the document image and transcribes the document into textual and non-textual components. OCR systems are capable of identifying words, characters, and sentences from document images (Ahmed et al., 2017). To the handwritten text and printed text the OCR system substantially reduce the time required for the conversion of a document to computer-readable form compared to the time required to enter the data manually by a human (Salvi, 2014). OCR techniques are broadly used in detecting and examining signatures at banks, converting books and documents into electronic files, or publishing the text on a web-site (Erkilinc et al., 2011).

To explore or write the document contents made possible by OCR, the output of the OCR system must contain uni-codes plus formatting information i.e., either ASCII or UTF- 8. These codes are easily rendered on any text-browser. Further, the result also contains the location information about the textual block. Such information is required to display the contents as well as graphics components of the fed document image. In general, the most desirable output format is XML. As XML files not only contain textual data but also contain formatting information. Other formats could be readable PDF, PDF/A, DOC/ DOCX, XLS, text, etc.

Layout analysis is the main component of a DIA system and plays an important role in pre-processing. However, regarding the Pashto language, the document images are not explored so far. The reasons are language-specific complexities, which may comprise; writing direction, unavailability of documents that contain Pashto text and graphics per document, multi scripts per image, language-specific layouts, etc. For example, one such layout contains the Pashto text and graphics. Figure 2 shows two documents that contain the Pashto text and graphics. The traditional OCR systems, designed for Pashto scanned images, cannot be directly applied to camera-captured images until the recognition is achieved for different layouts and patterns.

Figure 2 Pashto document images containing pashto text and graphics.

Hence, we focused on research regarding layout analysis, bringing into consideration the detection of Pashto text and graphics per document. More precisely, we developed a system that separates textual segments and graphics segments. If these segments are not separated/ classified before feeding them as input to OCR, the OCRs are incompetent to recognize the text along with graphics and leads to significant errors in the recognition. Therefore, we developed such a system that classifies Pashto text and graphics in the Pashto documents. Rest of the article is organized as follows. “Related Work” presents the related work about DIA system. “Dataset Creation” and “Proposed Methodology” discuss how the dataset is created and the proposed methodology for our experiments. “Experiment” and “Results and Discussions” present the experimental setup, results and discussion. “Conclusion and Future Work” presents the conclusion and future directions for the research.

Related work

There is enough work concerning the field of DIA systems and it has progressed more quickly in the late 1980s and early 1990s (Manwatkar & Singh, 2015). To narrow down the research space, we only report the related work that is more focused on the Pashto cursive language in the area of the DIA system. This section mainly covers the most relevant work concerning the layout analysis regarding cursive language.

Layout analysis and classification

O’Gorman & Kasturi (1995) introduced the document spectrum, for structural page layout analysis. The introduced approach is based on the bottom-up, nearest-neighbor clustering of page sections. The developed approach results in an absolute determination of skew, between-line spacing, and within-line, and finds text blocks and text lines. This method is more beneficial across several other methods in three basic aspects, (1) fairness from different text spacing, (2) freedom of a skew edge (3) the processing capability on confined areas of several text adjustments in the corresponding image.

Simon, Pret & Johnson (1997) developed a new method for document layout analysis based on bottom-up approach. Their algorithm’s implementation was done in the CLiDE (Chemical Literature Data Extraction) system (http://chem.leeds.ac.uk/ICAMS/CliDE.html). Their method is based on Kruskal’s algorithm. To create the physical page formation they used a special distance-metric among the segments. The achieved efficacy of the method by considering the 98 test images with an error rate of 1%.

Breuel (2003) presented some novel algorithms and statistical methods for layout analysis. They have assessed their method/approach on a subset of the UW3 (University of Washington Database3) database. Their algorithm delivers better performance regarding layout. Laven, Leishman & Roweis (2005) examined the effectiveness of analytical model recognition algorithms for determining the difficulties of physical and logical layout analysis. They proposed these problems in the quality of heuristic, grammar-based, and rule-based, approaches. and grammar-based techniques. They have developed a dataset (http://jmlr.csail.mit.edu) which contains 932 page images from academic journals. Finally, they applied three statistical classifiers to the logical layout analysis problem. They obtained an average precision of 85.5% among 16 sections and a general efficiency of 86%.

Antonacopoulos, Karatzas & Bridson (2006) proposed and discussed several essential problems circling the ground truth toward the performance evaluation regarding layout analysis techniques. They converged on some various steps like formation, description and creation steps in the setting of a unique dataset formed by the authors. The current dataset (http://www.prima.cse.salford.ac.uk/dataset) has been made freely available to researchers. Shafait (2008) modified the Breuel (Roman script text-line model) (Breuel, 2001) system for layout analysis to Nastaliq script, and formed a text line pattern for Urdu text-lines. For the evaluation performance of the specified layout analysis system, they scanned 25 Urdu documents from various sources, which were classified into five ( 5) classes i.e., magazine, book, poetry, digest, book, poetry, and newspaper. In the dataset, there are five images of each class. Their dataset (http://www.iupr.org/demosdownloads) has been made publicly available. The produced system was examined with data from chosen classes, and 92% precision was achieved for book, magazine, and poetry documents due to approximately extensive inter-line spacing. Despite this, the accuracy reduces to 80% for digest document level through small inter-line spacing and appearance of itemized lists, and for newspapers, the correctness falls up to 72% due to many font sizes, inverted text, and defective quality of the page.

Smith (2009) introduced a different hybrid page layout analysis algorithm. The stated algorithm used bottom-up techniques to create an initial data-type position and find the tab-stops that remained applied while the page was formatted. For the detection of the column layout of the page. They used tab-stops to force formation including a reading sequence on identifying regions, and the column layout was applied in a top-down manner. Their achieved accuracy rate of the developed method is 92%. The complete source code for their proposed work is freely available (https://github.com/tesseract-ocr) as part of the Tesseract.

Bukhari, Shafait & Breuel (2011) introduced a method for the layout analysis of Arabic and Urdu document images. These images comprised of a mixture of separate and multi-column layouts. They used a proper combination of well-accepted, hardy text and non-text segmentation, text-line extraction, and reading sequence perception methods in the given layout analysis system. They assessed the presented system on 20 Urdu and 25 Arabic document images having different layouts. They achieved accuracy toward textual data and non-text segmentation was 99% for the Arabic dataset. For text-line extraction, they used ridge based method and achieved accuracy of 96% for the Arabic dataset while 92% for the Urdu dataset. Erkilinc et al. (2011) introduced an algorithm for page layout analysis and document classification. In their proposed algorithm photo, text, and line/solid boundary areas classified. To improve and evaluate the proposed technique different varieties of easy, difficult, color, and gray-scale documents were applied. They made use of a text detection module and Run Length Encoding technique (RLE) (Xu et al., 2004), and a second module to identify photo/image and illustrated sections in that input document. Behind that, to get out photo candidate domains they worked a block-based classifier applying basis vector predictions. By using the Hough transform and edge-linkages analysis, the ultimate module recognizes lines and sharp edges sequentially. The photo, text, and strong edge/line maps are composed to create a page layout analysis of the scanned objective document. The investigation outcome shows 85% accuracy on average for the introduced method.

Tran, Na & Kim (2015) proposed a method for textual and non-textual segments classification in document images. Their introduced method is the combination of whitespace analysis method with multi-layer homogeneous regions which are called recursive filters. Empirical outcomes on the ICDAR 2009 page segmentation competition dataset and other datasets illustrate the effectiveness and advantage of the intended approach. They obtained above 90% efficiency for text detection, no-text detection, text region detection, and non-text region detection. Ahmad et al. (2017) presented a text-line extraction method toward the extraction regarding large titles and headings in Arabic like scripts. The proposed method is based on a Hanning window smoothing procedure and Horizontal Projection Pro-file (HPP) (Javed, Nagabhushan & Chaudhuri, 2014). They obtained an accuracy of 99.30%. The weakness of their introduced approach is that it requires d-skewed images.

Dataset creation

This section presents a sketch of the formulation of a real Pashto dataset in the area of document layout analysis and classification. Firstly, it shows the motivation for dataset creation. Secondly, it shows necessary information concerning image acquisition in the scope of layout analysis. Moreover, we report the unconstrained atmosphere for the camera captured acquisition. Plus, it also defines the approach of how we annotated/ transcribed and named the images.

Motivation

It is a valid fact that the synthetic data does not meet all the real-world requirements that could cover the real challenges of layout analysis and classification. The classifiers trained on synthetic data have a deficiency in generalization towards real data. Hence, efficient investigation mainly relies on covering real data rather than synthetic data. But, concerning the Pashto language, we do not have any such real data that can be analyzed for document layout analysis. Accordingly, we need to produce a real dataset that comprises primary patterns/ layouts and other complexities present in the Pashto language. To accomplish this purpose, we have created a new dataset that contains real-world samples of camera captured images presenting an appropriate benchmark for the Pashto DIA system. Some representations of Pashto text and graphics are shown in Fig. 3. Certainly, this research not only provides the scientific domain but also benefits the research society regarding regional languages. On the other-side real data shows more challenges due to the formation of various stages in the process.

Figure 3 Samples of images from our newly created dataset.

Further, to accomplish useful document layout analysis and classification, we require a suitable dataset toward our newly developed systems and models. Hence, the purpose of this section is to report the creation of a real dataset for the Pashto language in the domain of document layout analysis and classification.

Data acquisition

Data acquisition remains an essential stage in the dataset creation. The sources concerning data acquisition must be a scanner or a camera-based device. In usual, the scanner-based acquisition is admitted as a conventional method and is the most applied method towards the digitization of books and other historical stuff. Although, the scanner-based document is examined to be pretty simple due to their acquisition in a restrained condition. Moreover, you have to have an extra device named scanner. In contrast, modern growth in mobile technologies in the formation of smartphone gadgets, etc., the camera-based data acquisition brings notable concentration towards digitization.

It is to consider here that camera-based capturing frequently undergo due to artifacts like shadows, skew, perspective distortion, warping, light reflection, and blurriness. While the scanner based acquisition hardly presents skew and lack of different artifacts as compared with a camera-based acquisition. It indicates a classifier trained on camera-captured images will hold the strength to be extra robust associated to the one trained on scanned documents. Meanwhile in our research, we use a camera-based image acquisition procedure. We operated a mobile camera to captured the images of the chosen Pashto documents.

Dataset description

Our newly created dataset contain 1,017 Pashto document images captured by the camera. We named our dataset Camera Captured Pashto Document Image (CCPDI). The sources (https://www.taleem360.com/categories/text-books-kpk?page=2) for acquired books for the CCPDI dataset given in Table 1. To annotated the CCPDI dataset, we worked on a tool called labelMe. The annotation of every textual and graphics segment is made by analyzing its contour/edges by taking polygons. The concern annotation toward every image is saved in a separate .json file.

Table 1 Sources of acquired books for CCPDI dataset.

S. No	Name of book	Total pages	Pages acquired	
1	Wyaarh	400	137	
2	Pashto school text books	…	880	
	Total	…	1,017	

Data annotation

In supervised learning, the learning is achieved by contributed input-output pairs. It means for each input we have to provide a respective class label (Hussain et al., 2022). In short, we have to transcribed/ annotated or make ground truth for our data. Accordingly, annotation is the method in which we give a label to data like text, videos, and image, (our case). The Fig. 4 presents the input labeled image with the corresponding ground-truth. This is performed by specifying some kind of keywords in the appropriate area of text, image, etc. Normally, this approach is used in the scope of AI and ML areas to train the machines. A necessary step in improving any computer vision model is to establish a training algorithm and approve certain models applying high-quality training data.

Figure 4 Input image (A and B) the corresponding ground-truth.

Proposed methodology

To accomplish the intentions and to examine the layout analysis phase of the DIA system in the Pashto language, we applied the CNN with a deep learning approach. The model is based on the evolution of the sophisticated and classical variant of Faster R-CNN called Single-Shot Detector (SSD).

This alternative of the Neural network accepts the algorithms/methods to identify the problem including the parameter settings most suitable for the proposed task. We have given the focus on the main challenges that are to be faced in the detection of text and graphics in document images. First, the network was trained for this process, the initial weights were taken randomly and then, the training or learning started. First, we will briefly describe the region-based object detectors (like RCNN, FastRCNN, FasterRCNN). After this brief discussion, we discussed the variant of Faster R-CNN i.e., Single-shot detector (SSD) in detail in “Experiment” that we followed with a deep learning approach. Finally, we discussed the evaluation criteria for the measurement of the accuracy of the developed system.

Neural network

A neural network also called an artificial neural network (ANN). The system of hardware/ software imitated after the performance of neurons in the animal brain. This is one method for formulating artificially intelligent programs. Neural networks are a model of machine learning, where a program can change as its weight learns to solve a problem.

Recurrent neural network

RNN is a kind of neural network which is usually practiced in speech recognition and natural language processing (NLP). The output of the earlier stage is served as input to the present step. In conventional neural networks, all the inputs and outputs are independent of each other, particularly in situations like when it is required to divine the subsequent word of a sentence, the first words are needed and hence there is a requirement to retain the earlier words.

Convolution

To convolve means to roll together. The term convolution leads to the mathematical succession of two functions to form a third function. It joins two sets of information. In the situation of a convolutional neural network (CNN), the convolution is conducted on the input data including the usage of a kernel or filter (these words are applied mutually) to then create a feature map. Convolution and the convolutional layers are the central structure segments used in CNNs.

Convolutional neural network

A CNN is a kind of artificial neural network which is usually applied in image processing and recognition that is specially created to treat pixel data. CNN is one of the central classifications to do face recognition, object detection, image recognition, image classifications, etc., which are some of the domains where CNNs are broadly used. CNN applies deep learning to accomplish both generative and identifying jobs, usually applying machine vision that holds the image and video recognition, simultaneously with recommender systems and natural language processing (NLP).

Object detector: region-based object detector

Region based convolutional neural network

R-CNN stands for “Region-based Convolutional Neural Networks”. R-CNN, with the R standing for the region, is for object detection. Alternatively, of operating on an extensive quantity of regions, the RCNN algorithm suggests a collection of boxes in the image and tests if either these boxes include any object. RCNN practices a particular exploration to remove these boxes from an image. R-CNN aims to get in an image, and accurately recognize where the central objects in the image (Parthasarathy, 2017; Zhang et al., 2020; Wang et al., 2021).

Fast R-CNN

To handle the drawbacks of R-CNN a faster object detection algorithm was introduced and it is known as Fast R-CNN. The procedure is comparable to the R-CNN algorithm, however, rather than feeding the region proposals to the CNN, we give the input image to the CNN to create a convolutional feature map. The Fast R-CNN is faster than R-CNN is because we do not have to serve 2000 region proposals to the convolutional neural network at any time. Alternatively, the convolution process is performed just once per image and a feature map is generated from it. Fast R-CNN is executed in Python and C++ (using Caffe) and is available following the open-source MIT License at http://github.com/rbgirshick/fast-rcnn (Girshick, 2015).

The single shot detector

The single shot detector (SSD) is a variant of Faster R-CNN. It has no designated region proposal network. SSD divines the boundary boxes and the classes direct from feature maps in one single shot. Through our research purpose, we worked mainly on SSD (Liu et al., 2016) model. SSD is created for object detection in real-time. Faster R-CNN works on region proposal network to form boundary boxes and applies those boxes to classify objects. It is considered the state of the art in accuracy, the whole process runs at seven frames per second. SSD speeds up the process by decreasing the requirement for the region proposal network. To improve the drop in accuracy, SSD works many enhancements including multi-scale features and default boxes. These improvements enable SSD to match the Faster R-CNN’s efficiency applying lower resolution images, which further accelerates the speed higher (Liu et al., 2016).

SSD model

The SSD method provides a limited-size number of bounding boxes and rates concerning the appearance of object class occurrences in those boxes, served by a non-maximum suppression-step to perform the final detection. The initial network layers are based on a standard architecture that trained for high-quality image analysis, which we call the base network. Further, the auxiliary structure added to the network to achieve detection with the following key features (Liu et al., 2016). Multi-scale feature maps for detection.

Convolutional predictors for detection.

Default boxes and aspect ratios.

The SSD object detection is composed of two parts. First extract feature maps and then apply convolution filters to detect objects.

The SSD works VGG161 to extract feature maps, and next, it detects objects applying the Convolution in layers 4−3. The SSD architecture is trained on more than a million images of the ImageNet database.2 It calculates both the location and class scores utilizing small convolution filters. After extracting the feature maps, SSD uses 3×3 Convolutional filters for every cell to perform predictions. Certain filters determine the results just like the regular CNN filters. The typical SSD model is shown in Fig. 5.

Unlike its predecessor R-CNN (Girshick, 2015), the SSD does not use a selected region proposal network, instead, it divines the boundary boxes toward the classes direct from feature maps in a single pass. This network is renowned for speed and performance regarding object detection problems. SSD uses the VGG16 network as a feature extractor. Next combine custom convolution layers (blue) afterward and apply convolution filters (green) to make predictions (Girshick, 2015; Liu et al., 2016).

Figure 5 A typical single shot detector model (Liu et al., 2016).

However, convolution layers decrease spatial dimension and resolution. Therefore, Fig. 6 shows the detection of large objects only. To fix that, we perform autonomous object detentions from multiple feature maps. Figure 7 shows the multi-scale feature map for detection. The dimensions of feature maps is shown in Fig. 8. SSD uses deep-down layers to the convolutional network for the detection of objects. If we redraw the picture closer to scale, we should understand that spatial resolution has decreased significantly and may already drop the chance in locating small objects that are very difficult to detect in less resolution. If such difficulty exists, we require to enhance the resolution of the input image (Girshick, 2015; Canziani, Paszke & Culurciello, 2016; Liu et al., 2016).

Figure 6 Prediction for location and classification (Liu et al., 2016).

Figure 7 Feature map for detection (Liu et al., 2016).

Figure 8 SSD diagram showing the dimensions of feature maps (Liu et al., 2016).

Training SSD

The major contrast among training SSD and training a typical detector that utilizes region proposals is that ground truth information needs to be allocated to appropriate outputs in the established set of detector outputs. Once this responsibility is defined, the loss function and backpropagation are applied end-to-end. Training also includes collecting the set of default boxes and scales for detection as well as the strong negative opening and data enlargement procedures (Liu et al., 2016).

Evaluation criteria

We will use mAP (mean average precision) and Intersection over Union (IoU) for the evaluation of our proposed model. The mentioned metrics are briefly explained below.

mAP (mean average precision)

To measure accuracy, we use average precision (AP), which is a popular metric regarding measuring the accuracy of object detectors. AP measures the average precision value toward recall value above 0 to 1 (Everingham et al., 2014). First, the AP is calculated for every class and next averaged over several classes. The mAP3 (RADAR: AI Edition, 2018) for object detection is the average of the AP calculated for all the classes. It seems complex but much easy as we illustrate it with an example. But before that, we will take a sharp recap on precision, recall, and IoU.

Precision and recall

Precision measures how much the prediction is efficient. i.e., the percentage of predictions are accurate.

Recall measures how useful you achieve all the positives. The mAP can be calculated via Eq. (1).

(1) mAP=∑q=1QavP(q)Q

where avP (q) is the average precision (AP) for a given query (q) and ‘Q’ is the total number of queries.

IoU

IoU measures the two overlap boundaries. IoU is used to measure how much the ground truth bounding-box overlaps with the predicted boundaries (Everingham et al., 2014). IoU can be measured via Eq. (2).

(2) IoU=Ag∩ApAg∪Ap

where Ag refers to the area of bounding box obtained from ground truth, while Ap refers the area of bounding box obtained from predicted coordinates.

Experiment

In this section, we explained the splitting of data as training data and testing data. The object detection API and the supportive libraries of the API have also been presented. In last the entire discussion about the experiment also been performed in detail. The training of the model and exporting the learned model is also explained.

Data split

In supervised learning, proper train sets, as well as test sets, are extremely essential for achieving robust classifiers. For this purpose, we have split the dataset (shown in “Experiment”). We have worked a hold out method, in which about 70% of data belongs to train-set and about 30% goes to test-set. We particularly used the same proportions. However, before utilizing this split, we have mix-up the entire dataset first. The total 1,017 images are split in a ratio of 71:29. We could get a train-set of 71% images and a test-set of 294% images. The corresponding ground-truth data are similarly split according to the images.

Experimental procedure

The empirical procedure has been divided into two sub tasks. The first task is Training the model and the other task is exporting the model. To accomplish these two sub tasks the split data has been converted to TFRecord file format. TFRecord file is the combination of the ground-truth and the corresponding image file formats.

Training the model

After the creation of the input file for API, the selected model has been trained by fulfilling the following specifications. An object detection training pipeline. They also provide sample configuration files on the repository. The SSD-mobilenet-v1-config has been utilized as a basis in my training. The need for the adjustment of the num-classes to two (Pashtotext, graphic) has been covered. I worked the default settings in terms of other configurations like the learning rate 0.003, batch size 24.

The dataset (TFRecord files) and its corresponding label map. The label map file includes the class names, it is needed to the classifier to learn textual blocks corresponding to class labels. In our case, the file has only two class names i.e., “Pashtotext” and “graphic”. Figure 9 also shows the snapshot of the label-map file.

Pre-trained model checkpoint. A pre-trained model is used to train the API for our own task. Because for limited data a completely un-trained model will take large time to converge. This process is also known as re-tuning or fine-tuning an already available models. We use SSD-mobilenet-v1, which is already trained on Pet dataset. A pre-tuned model the SSD-mobilenet-v1 as this model speed is more important than accuracy has been used.

Figure 9 Class labels in a label map file.

The training process has been performed locally. While training the evaluation job also done after every 100 epochs. By running Tensor-Board on our machine the training and evaluation process has also been monitored. The training ran over 29.75k steps with a batch size of 24 and achieved good results in about 11h.

Exporting the model

After completing the training phase, I exported the trained model to a single file that is generally known as a frozen inference file.

I created the model checkpoints on my local machine after running the path specified to the trained classifier and then used the provided script to export the model.

Machine platform

The experiments have been carried out using a machine having the following hardware configuration; Intel Core (TM)i7, 2.8GHz, with 16 GB RAM. The machine has a GPU of Nvidia GTX 1,070 chip, with 8 GB internal memory.

Results and discussions

In this section, we explained the splitting of data as training data and testing data. The object detection API and the supportive libraries of the API have also been presented. In last the entire discussion about the experiment also been performed in detail. The training of the model and exporting the learned model is also explained.

Results

After 29,750 epochs, we have stopped the training, and the final model was selected for evaluation. The evaluation was done by examining the 300 images from the test set. In this way, We achieved a mean average precision (mAP) 84.90%. Figure 10 shows the mAP in detail along with corresponding epochs as well as total elapsed time.

Figure 10 The overall evaluation that is carried out along with the training process.

The final mAP is 84.90%.

Similarly, the training process was tested for how it reduces the total loss. The total loss that we finally achieved is 4.6, while its ceilings were approximately 8.9. Figure 11 depicts the value of loss with regard to epochs in the training.

Figure 11 The loss was gradually reduced from 8 to 4.6 after 29,750 epochs.

Discussion

After an accurate analysis of results as well as individual images, we have important findings that are described in the following lines. Before we could empirically asses the results, Fig. 12 shows some prediction of our proposed model on totally unseen data. To know the visual description of our examples, please understand the notions mentioned here. The light green color for the bounding box represents “Pashto text” and the light blue color represents “graphic”. The first row in the Fig. 12 represents the two cases where our model has shown the best performance. The image at left shows the prediction and the image at right shows the ground-truth.

Figure 12 Some example of testing our proposed model, (A and B) belong to easy performance, (C and D) are of moderate complexity while (E and F) more complex images.

On the other hand, the middle row describes images with less complexity which is shown in Fig. 12. The main cause is the quality of the bounding boxes, for example, some bounding-boxes partially bound Pashto text as well as the graphic. In general, our proposed method has proven better performance while checking it via human eyes.

Although, the final row in Fig. 12 presents some images with less detection and miss-classification. I thoroughly examine the reason, and I found that for small bounding-boxes the prediction is not good as for large bounding-boxes. Additional investigation yields that the instances for the small textual blocks are similarly less than the large textual blocks. This non-uniform distribution affects the classifier to determine more about large textual blocks rather than small textual blocks. I further validate this point by specifically considering the mAP for only small bounding-boxes, and therefore it is proved there that the mAP for small bounding boxes is just 25%. It can be observed in Fig. 12.

Conclusion and future work

This research for the first time presents a pioneering study concerning the layout analysis and classification of Pashto document images. The research especially examined the classification of Pashto text and graphic in Pashto document images. For this purpose, we have first formed a new dataset that comprises real Pashto document images. The images are taken via a handheld camera. The dataset is freely available and will be a significant resource for the research community for analyzing the DIA domain in cursive scripts.

Moreover, this research proceeds one step ahead and apply the deep learning-based method to examine how we can detect/ classify Pashto text and graphic in a single document image. We have preferred the SSD model that has a hybrid model containing VGG16 as convolutional layers and a neural network for learning high distinctive features. Concerning the Pashto language, this work is the first work that using deep learning-based architecture and examines the Layout and Classification stage of a DIA system. We have achieved an overall mAP of 84.902% as a benchmark. The results are ensuring as it is the first-ever effort to analyze and investigate the Pashto language with regard to the DIA system.

We observe that textual/graphical blocks which are comparatively small or bounded in a small bounding-box shown poor results. We require more research to empirically investigate this issue and find out the real causes that lead us to such poor results. The reasons could be the small size of the dataset or the fewer instances especially compared to small bounding-boxes. Similarly, we used rectangular bounding-boxes to make ground-truth information for training. The rectangular bounding-boxes are very much suitable for regular shapes while textual blocks are mainly irregular. In brief, polygon-based labeling will be another dimension to explore as future work.

Additional Information and Declarations

Competing Interests

Author Contributions

Data Availability

1 VGG16 is the creation of Visual Geometry Group (VGG) and contains 16 convolutional layers.

2 The ImageNet project is a large visual database designed for the recognition of visual objects. The database consists of more than 14 million images (Deng et al., 2009).

3 A Beginner’s Guide to Object Detection.

The authors declare that they have no competing interests.

Khan Bahadar conceived and designed the experiments, performed the experiments, analyzed the data, performed the computation work, prepared figures and/or tables, authored or reviewed drafts of the article, and approved the final draft.

Riaz Ahmad conceived and designed the experiments, performed the experiments, analyzed the data, performed the computation work, prepared figures and/or tables, authored or reviewed drafts of the article, and approved the final draft.

Khursheed Aurangzeb conceived and designed the experiments, performed the experiments, analyzed the data, performed the computation work, prepared figures and/or tables, authored or reviewed drafts of the article, and approved the final draft.

Siraj Muhammad conceived and designed the experiments, performed the experiments, analyzed the data, performed the computation work, prepared figures and/or tables, authored or reviewed drafts of the article, and approved the final draft.

Khalil Ullah conceived and designed the experiments, performed the experiments, analyzed the data, performed the computation work, prepared figures and/or tables, authored or reviewed drafts of the article, and approved the final draft.

Ibrar Hussain conceived and designed the experiments, performed the experiments, analyzed the data, performed the computation work, prepared figures and/or tables, authored or reviewed drafts of the article, and approved the final draft.

Ikram Syed conceived and designed the experiments, performed the experiments, analyzed the data, performed the computation work, prepared figures and/or tables, authored or reviewed drafts of the article, and approved the final draft.

Muhammad Shahid Anwar conceived and designed the experiments, performed the experiments, analyzed the data, performed the computation work, prepared figures and/or tables, authored or reviewed drafts of the article, and approved the final draft.

The following information was supplied regarding data availability:

Ibrar Hussain. (2024). adqecsbbu/Pashto-Text-Graphic: v1.0.0. Zenodo. https://doi.org/10.5281/zenodo.10569204.

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
