# Peer review of "Pashto script and graphics detection in camera captured Pashto document images using deep learning model"

_PeerJ Computer Science, doi:10.7717/peerj-cs.2089_

## Round 0.1 · original submission · Major Revisions

· Academic Editor

Major Revisions

See comments from three reviewers.

Reviewer 1 ·

Basic reporting

In general, the paper is presented well. but the Link of the dataset, literature review may be revisit for improvement if possible. Carefully proofread the entire document to correct any grammatical errors, or typos. This step is crucial for ensuring a polished final product. seeking input from a language editing service or a colleague proficient in English may further refine the paper before submission.

Experimental design

no comment

Validity of the findings

How this model are best fit for your case. In general as this model will detect graphic component in other scripts (English, Chinese, Arabic, Hindi). What research gap were filled by your work. Explain the need of dataset for this research work. The author need to answer the following questions.
Difference of Camera vs Scan images, on model.
If possible validate RESULTS with other model deep learning model
Scan image are better for your work or only camera captured. Do you ever think that this might limit the scope of work? Yes/No
If the last question answer is No, then justify and state for OCR application the difference of scan images and Camera images.

Reviewer 2 ·

Basic reporting

no comment

Experimental design

no comment

Validity of the findings

no comment

Additional comments

Overall, the paper under review titled "Pashto script and graphic detection in camera-captured Pashto document images using deep learning model" is presented well but the authors need to address a few of the concerns to improve the quality of the manuscript.
As the authors claimed that this work lies in the field of OCR, Please highlight OCR-related contents in the Introduction section.
Explain the Detection mechanism do you use for extraction of Text vs Pictures or recognition, as OCR mainly recognizes text, put some detail on how detection is used here.
Another important question is why Camera Capture is in the Title, so if Scanner then the result will be different?
What were the data sources, also, consider ethical implications and ensure that the dataset collection and model development may not violate the privacy of humans, race, culture religions, etc.
What is the F1 score for both script and graphic detection?
Improve the quality of images.

Reviewer 3 ·

Basic reporting

The paper in general well structural and organized and we’ll be more refine if the following improvement are added.
Related work may be up to dated, may be checked for addition if any

Please revise all Tables and Figures. Some figures are not clear, so kindly upload high quality images, e.g. Figure 10, and 13. Figure 9 is not necessary.

Ensure that the references are complete, accurate, and follow a consistent citation style.
What is the main contribution of this paper to the available literature domain?

Experimental design

The result may be check in a comparison with the model used in one of the following published papers. The author may read the following paper for read reference to compare their result with one or more model as used in the following paper.

Wang, N., Liu, H., Li, Y., Zhou, W., & Ding, M. (2023). Segmentation and Phenotype Calculation of Rapeseed Pods Based on YOLO v8 and Mask R-Convolution Neural Networks. Plants, 12(18), 3328.
Chen, W., Lu, S., Liu, B., Chen, M., Li, G., & Qian, T. (2022). CitrusYOLO: a algorithm for citrus detection under orchard environment based on YOLOV4. Multimedia Tools and Applications, 81(22), 31363-31389.
Biró, A., Jánosi-Rancz, K. T., Szilágyi, L., Cuesta-Vargas, A. I., Martín-Martín, J., & Szilágyi, S. M. (2022). Visual Object Detection with DETR to Support Video-Diagnosis Using Conference Tools. Applied Sciences, 12(12), 5977.

Validity of the findings

.

---

## Round 0.2 · accepted · Accept

· Academic Editor

Accept

The revision answered the comments. Accept

Reviewer 1 ·

Basic reporting

The Authors addressed all the concerns raised in my previous review. Upon reviewing the revised manuscript, I find it to be a significantly improved version. The content is well-researched, and the arguments are well-supported. I am pleased to recommend its acceptance with some minor suggestions.
•⁠ ⁠Grammatical and Formatting Issues: There are a few minor grammatical mistakes throughout the paper, and I noticed some formatting inconsistencies, particularly in the reference section. Please review the references and ensure they conform to the journal's format guidelines.
•⁠ ⁠Figure 5: The clarity of Figure 5 is not optimal. It appears to be of low quality, making it difficult to discern important details. Kindly provide a high-quality version of Figure 5 to enhance its readability and effectiveness.
Once these minor changes are addressed, I believe the paper will be ready for publication.

Experimental design

No problem with the experimental design.

Validity of the findings

Findings are valid.

Additional comments

No comments

Reviewer 2 ·

Basic reporting

no comment

Experimental design

no comment

Validity of the findings

no comment

Additional comments

Authors of the manuscript titled "Pashto Script and Graphic Detection in Camera Captured Pashto Document Images Using Deep Learning Model" have effectively addressed the issues raised in my previous comments. Therefore, I am inclined to recommend its acceptance. One minor revision is suggested:

While the clarity of the manuscript has significantly improved, Figure 5 and Figure 8 remain unclear. I recommend the authors upload a high-quality version of the Figures to ensure clarity and enhance the effectiveness in conveying essential details